# Identification of Fibrillarin and Cajal Bodies Under DNA Replication Stress Conditions in Root Meristem Cells of *Allium cepa*

**DOI:** 10.3390/ijms262311321

**Published:** 2025-11-23

**Authors:** Aneta Żabka, Natalia Gocek-Szczurtek, Mateusz Wróblewski, Justyna Teresa Polit

**Affiliations:** Department of Cytophysiology, Faculty of Biology and Environmental Protection, University of Lodz, 90-236 Lodz, Poland; natalia.gocek@edu.uni.lodz.pl (N.G.-S.); mateusz.wroblewski@biol.uni.lodz.pl (M.W.); justyna.polit@biol.uni.lodz.pl (J.T.P.)

**Keywords:** onion root meristems, DNA replication, hydroxyurea, nucleolus, RNA transcription

## Abstract

The correct course of DNA replication is crucial to maintaining the integrity of the genome. Any abnormality in this process inevitably leads to replication stress (RS). Hydroxyurea (HU) is a replication stressor widely used to inhibit DNA biosynthesis by depleting the deoxyribonucleoside triphosphate (dNTP) pool. The aim of the study was to examine how the 24-, 48-, and 72 h exposures to 0.75 mM HU affect the localization of fibrillarin (FBL; a highly conserved nucleolar protein and the component of Cajal bodies) and the amount of rRNA transcripts (detected using 5-ethynyl uridine; 5-EU), in root meristem cells of *Allium cepa*. The consequence of prolonged RS was initially (after 24 h of incubation in HU) a 2-fold increase in 5-EU incorporation into the nucleolus, then (after 48- and 72 h incubations) followed by a gradual decrease in rRNA transcription to a level similar to that of the control. In interphase and in early prophase, both in the control material and during successive periods of incubation of root meristems in HU, the immunofluorescence of FBL accumulated in the fibrillar centers (FCs) of the nucleoli, in the dense fibrillar components (DFC), and in the granular components (GC). In some HU-treated metaphase cells, FBL was localized around the telomeres of the chromosomes, while in telophase, it was found in the fragmented chromosomes. In addition, an increase in the number of Cajal bodies (CBs) was observed during subsequent incubation periods with HU. After 48 and 72 h of treatment with HU, the number of CBs was found to be almost twice that observed in the control series. CBs disappeared in prophase and reappeared in interphase. These results suggest that depending on the duration of RS, changes in the level of rRNA transcription and in the abundance of CBs may correlate with the production of RNP and ribosome biogenesis.

## 1. Introduction

DNA replication is a fundamental process that occurs before each cell division and is based on the sequential activation of a large number of genes and the coordinated action of replication proteins [1]. DNA biosynthesis is often influenced by both endogenous and exogenous stress factors. Internal factors include reactive oxygen species (ROS), which are produced as the by-products of cellular metabolism, and reactive aldehydes generated as a result of alcohol metabolism, which may form cross-links in the DNA [2]. Exogenous DNA-damaging factors include ultraviolet (UV) and ionizing (IR) radiation, as well as various chemicals such as methylmethanesulfonate (MMS), mitomycin C, psoralen, cisplatin, etoposide, and camptothecin (CPT) [2]. A number of factors, such as pyrimidine dimers, DNA lesions, and the formation of DNA structures, can alter the process of DNA synthesis, epigenetic features of chromatin, and, consequently, may induce a so-called replication stress (RS) [3,4,5,6]. RS results in slowing down or inhibition of the replication fork (RFs) progression, and consequently, in their reversal or collapse [7,8]. Stalled RFs may collapse after prolonged inhibition involving the separation of the replisome complex from the DNA. Collapsed RFs are unable to restart replication, and dormant origins must be fired once more in order to complete DNA replication. The slow movement of RF and the resulting global arrest of origin firing increase the time required for DNA duplication. The end of the DNA synthesis phase (S-phase) must therefore be delayed to allow all DNA to be replicated before the onset of mitosis [9,10].

Proteins closely associated with DNA, such as histones, can also interfere with the movement of RFs. Other proteins, such as those forming the pre-Replicative Complex (pre-RC) in dormant origins localized in kinetochores or in centromeres, need to be tightly bound to DNA to maintain their function. This, however, may also affect the movement of RFs and cause topological stress [11]. In the case of RFs barriers, all engaged proteins are recruited to the DNA strands to deliberately inhibit the foreheads. These barriers are often unidirectional and prevent collisions between RFs and transcription vesicles. Replicative and transcriptional machinery moving along the same DNA template can lead to conflicts. The replication-transcriptional collisions are associated with the slowing down or stopping of RFs and with the formation of so-called R-loops (RNA:DNA hybrids), formed between RNA transcripts and a single strand of DNA [12,13]. These can impede the movement of RFs, expose sensitive ssDNA, and cause the formation of DNA double-strand breaks (DSBs) after transcription-coupled nucleotide excision repair [14]. To avoid such collisions, replication and transcription must be spatially and temporally coordinated in eukaryotic cells.

Hydroxyure (HU) is an inhibitor of ribonucleotide reductase (RNR), which converts ribonucleoside diphosphates (rNDPs) to deoxyribonucleoside diphosphates (dNDPs) using a tyrosyl free radical, usually in the iron-sulfur center of the enzyme [15]. Furthermore, HU has been shown to be a potent inhibitor of class I ribonucleotide reductases in prokaryotes and eukaryotes in vitro, resulting in a reduction in the pool of dNTPs necessary for DNA replication and repair [15,16]. Earlier studies have shown that 72 h incubation of 4-day-old primary roots of *A. cepa* [17,18,19,20,21] and *Vicia faba* [22] with 0.75 mM HU results in the disruption of the S-M (DNA synthesis-mitosis) checkpoint function, leading cells either to premature chromosome condensation (PCC, also known as premature mitosis) [23] or to asynchronous nuclear chromatin condensation, simultaneously forming interphase (I) and mitotic (M) domains at opposite poles of the nuclei (IM cells). In addition, it has been shown that there is an association between advanced M poles of IM cell nuclei and polarized accumulation sites of auxin efflux carriers (PIN2 proteins) and IAA (indole-3-acetic acid) [20]. The asymmetric increase in IAA levels in the PIN2-enriched region of the cell decreases the inhibitory functions of the inhibitors of the CDK/kip-related Proteins (ICKs/KRPs) and, consequently, enables the activation of CDK–CBL complexes [24] at one side of the cell, the M-pole, as opposed to the other side, the I-pole. At the same time, such observations seem to be consistent with studies by Himanen et al. [25] and Vanneste et al. [26] showing the auxin-dependent reduction in the activity of the KIP/CIP family of Cdk inhibitors.

*Allium cepa* is an excellent subject for cytological studies because it has a high DNA content (33.4 pg) and large chromosomes, which are easy to observe under the microscope. To test our hypothesis that prolonged (24-, 48- and 72 h) treatment with 0.75 mM HU may contribute to the reduced transcription of pre-rRNA and to the enhanced nucleolar stress (NuS), we analyzed onion root meristem cells with respect of the level of transcription (by incorporation of 5-ethynyl uridine, 5-EU) and the location and the intensity of fibrillarin (FBL) immunofluorescence (staining using antibodies). Since Cajal bodies (CBs) are highly conserved sub-nuclear structures, functionally associated with the nucleolus, playing an important role in RNA metabolism and in the formation of ribonucleoprotein complexes (RNPs; involved in transcription, splicing, ribosome biogenesis, and telomere maintenance), the aim of our study was also to assess whether prolonged HU-induced RS affects the number of CBs.

## 2. Results

### 2.1. Replication Stress (RS) Generates Chromosomal Aberrations

Comparisons of microscopic images of the Feulgen-stained root meristem cells (Figure 1a) from the control onion seedlings (Figure 1b) and those from the HU-treated plants (Figure 1c–h) revealed distinct changes in mitotic chromosome architectures. In the latter case, the frequency of cells with chromosomal aberrations after 72 h of incubation with 0.75 mM HU increased to 15% (±0.8) at 72 h. In the early stages of anaphase, abnormalities manifested in the abnormal pulling of chromatids to opposite poles of the cell (Figure 1c,d). A clear sign of RS was the appearance of numerous remnants of chromosome fragments in the equatorial plane and the formation of anaphase-telophase bridges (Figure 1e,f). At telophase, lost chromosome fragments (Figure 1g) often contributed to the formation of abnormal post-telophase nuclei (Figure 1h).

### 2.2. Identification of Fibrillarin (FBL) in Onion Root Meristematic Cells

Fibrillarin (FBL) analyses were performed for the control and HU-treated onion root meristem cells obtained from seedlings after three successive incubation periods: 24, 48, and 72 h. These included: (i) mean intensity of FBL labeling and (ii) mean number of Cajal bodies (CBs) per cell. All analyses were performed for the G2 phase nuclei selected by microfluorimetric measurements of propidium iodide-stained nuclear DNA.

During mitosis, in prophase, FBL filled almost the entire interior of the nucleolus (Figure 2a). At metaphase (Figure 2b), when the nucleolar material diffused into the cytoplasm, dispersed FBL was found to accumulate at the periphery of the chromosomes and was observed there until anaphase (Figure 2c). At the end of mitosis, during telophase, FBL accumulated again in the nucleolus (Figure 2d). In onion root meristems treated with 0.75 mM HU, clear differences in the localization of FBL were observed during metaphase (Figure 2e) and telophase (Figure 2f,g). Only accidentally (statistically in two metaphase cells per root meristem), FBL did not diffuse but accumulated in large amounts around the telomere of the chromosome (Figure 2e). Furthermore, 72 h HU treatment, leading to marked aberrations in telophase, resulted in about 50% of telophase cells with FBL located in fragmented chromosomes (Figure 2f,g).

Immunofluorescence detection of FBL in interphase cells revealed bright green signals localized within the nucleoli (Figure 3a–d). FBL was found in the fibrillar centers (FCs), dense fibrillar components (DFC), and the granular component (GC; Figure 3a–d). It was observed that, compared to the control, the average intensity of FBL staining increased slightly after 24 and 48 h and then (at 72 h) returned to a level similar to the control (Figure 4). Fluorescence of FBL in the nuclei can be seen in the line graphs shown in Figure 3a’–d’ and in the interactive 3D surface plots (Figure 3a”–d”).

### 2.3. Cajal Bodies (CBs) in Root Meristem Cell Nuclei

CBs are small nuclear organelles located within the nucleus in all phases of interphase (Figure 5a–e). CBs disappear in prophase and reappear in late G1 after transcription has been resumed in the progeny nuclei. CB analyses performed for cell nuclei in the G2 phase showed that only after 24 h of incubation with HU, a statistically significant increase in the number of CBs (9.2 ± 2.3) can be observed, compared to the control group (4.9 ± 2.0; Figure 6). After 48 and 72 h of HU treatment, the number of CBs increased to approximately 11 (11.4 ± 2.4 for 24 h and 11.7 ± 2.8 for 48 h; Figure 6).

### 2.4. HU-Induced Changes in the Dynamics of Transcription

Total transcriptional activity was assessed in both control and HU-treated root meristem cells of *A. cepa* by incorporation of the alkyne-modified uridine analog 5-ethynyl uridine (5-EU). Using the Click-it reaction and fluorescence analysis, this method allows the detection of newly formed transcripts generated by RNA polymerases. The strongest EU signals were observed in the nucleoli, which were associated with intensive synthesis of ribosomal RNA (Figure 7a,c,e,g). A control experiment without EU revealed no immunofluorescence signals (Figure 7a’). Fluorescence intensity (FI) analyses were performed for the nucleolar region and extranucleolar chromatin within the cell nucleus (Figure 7i). Compared to root control meristem cells, 24 h treatment of plant roots with HU resulted in a 2-fold increase in the FI of the nucleolus. Subsequent hours of incubation brought about a gradual decrease in fluorescence signals by up to a level similar to control cells (Figure 7i). However, in the case of EU signals observed in the extranuclear areas of cell nuclei, a 24 h HU treatment resulted in a string decrease in EU incorporation, compared with the control. Successive time points of HU treatment (48 and 72 h incubations) resulted in an increase in FI, but the fluorescence was noticed at still lower levels than in the control series (untreated root meristem cells (Figure 7i).

Strong fluorescence of the nucleolar regions, visible against the background of much less fluorescent areas of the extranucleolar chromatin, is shown in Figure 8a–d. The differences in fluorescence intensity of the extranucleolar chromatin and in the nucleolus can also be seen in the line (densitometric) plots (Figure 8a’–d’) and in the interactive 3D surface plots (Figure 8a”–d”).

## 3. Discussion

Ribosome biogenesis is a highly coordinated process closely linked to protein synthesis, cell proliferation, differentiation, and apoptosis [27]. Disruption of this process results in a phenomenon called nucleolar stress (NuS) [28]. In order to maintain homeostasis, molecular systems that respond to this stress are activated. Although it was initially thought that NuS is an outcome of abnormalities in rRNA processing or a result of modifications in ribosome assembly, increasing data suggest that many factors, such as DNA damage response or oncogenic stress, are also involved in signaling mechanisms leading to NuS. It is also known that RS, which is caused by the arrest and collapse of replication forks, blocks rRNA transcription and induces NuS [28].

The main nucleolar protein is fibrillarin (FBL; 33–36 kDa), associated with the C and D box-containing (box C/D) small nucleolar RNAs (snoRNAs) and involved in ribosome biogenesis, including pre-rRNA processing and 2′-O-ribose methylation of rRNA (2’-O-Me) and snRNAs (small nuclear RNAs) [29,30]. FBL is a conserved S-adenosyl-L-methionine-dependent methyltransferase found in all eukaryotic cells [31]. FBL was first detected in fibrillar and granular regions of the nucleolus in autoimmune serum of patients with scleroderma, as well as in Cajal bodies (CBs). FBL was found in NOR regions at metaphase and anaphase; during telophase, it was considered to be an early marker of the site of formation of the newly formed nucleolus [32]. In plants, FBL was first identified in onion cells within the transition zone between the fibrillar center (FC) and the dense fibrillar component (DFC) [33]. At least two different genes for FBL have been identified in most plants. In *A. thaliana*, both genes, encoding nearly identical proteins, were shown to have high levels of expression in cells characterized by an enhanced transcriptional activity. Fibrillarin 2 (AtFib2) shows a novel ribonuclease activity that is not observed for fibrillarin 1 (AtFib1). Moreover, another difference lies in the ability of both proteins to interact with phosphoinositides and phosphatidic acid [31]. There are three domains in both proteins: a glycine- and arginine-rich domain (GAR domain), a methyltransferase domain, and an alpha region [34]. The GAR domain is methylated at arginine residues and is responsible for interactions with various cellular proteins. The role of the GAR domain in nucleolar localization is not clear. Some studies suggest that this domain is not required for FBL targeting to the nucleolus or CBs, although it contains a nuclear localization signal (NLS); other results indicate that the GAR domain provides nucleolar targeting of FBL in plant and human cells [29].

Our research on onion root meristem cells shows that changes in the localization of FBL are correlated with the progression of the cell cycle. Using immunofluorescent detection of FBL during interphase and in early prophase (both in control seedlings and during successive periods of HU treatment), FBL accumulated in the FC, DFC, and GC of the nucleolus. During metaphase and anaphase, FBL diffused into the cytoplasm, and at the end of mitosis, it reassembled in the newly formed nucleoli. Although after HU treatment, the localization of FBL in interphase is the same as in HU-un-treated cells, clear differences were revealed during mitosis. In some metaphase cells, FBL was localized in large amounts around the terminal part of one of the chromosomes, while in telophase, FBL accumulated in fragmented parts of the chromosomes. This may indicate the involvement of FBL in NOR structures during mitosis.

CBs are small nuclear organelles, lacking a membrane, which can divide into daughter bodies or undergo fusion [35,36]. They are involved in RNA transcription and editing, such as mRNA splicing and modifications of small nuclear/nucleolar ribonucleoproteins (sn/snoRNP) [29,37]. Despite this multifunctionality, CBs are generally nonessential for cell survival in either plant or animal organisms [38]. However, in certain species, e.g., during embryogenesis in mice and zebrafish, functional CBs are needed for the completion of the developmental processes and cell survival [39,40]. CBs are thought to effectively connect various cellular components in an organized manner, acting as factories to increase the speed of cytophysiological mechanisms [36]. The experiments on the hypothalamic magnocellular neurons of supraoptic nuclei (SON) have shown that 24 h after injection of hypertonic saline, there is a 3-fold increase in the number of CBs, suggesting a rapid response to changes in transcription rate [41]. In our work, CBs localized within the extranucleolar chromatin, disintegrated in prophase, and reappeared in G1 phase following resumption of transcription in the daughter cell nuclei. In onion root meristem cells, successive incubation periods with HU brought about an increase in the number of CBs, suggesting that their abundance may be associated with increased production of RNP (ribonucleoproteins), splicing, or ribosome biogenesis [42,43,44]. Therefore, it cannot be ruled out that CBs are transcription-dependent intranuclear “organelles” that respond to replication stress induced by HU.

In this study, transcriptional activity of cell nuclei was assessed in both control and HU-treated *Allium cepa* root meristems by incorporation of 5-EU, which allowed us to detect the newly formed transcripts generated by RNA polymerases. Compared to the control series, 24 h exposure of onion roots to HU resulted in a twofold increase in fluorescence intensity within the nucleolar region, which, according to Muhammad [45], might be caused by stress-induced hyperactivation of rDNA. However, subsequent hours of incubation in HU brought about a gradual decrease in rRNA biosynthesis to levels similar to those observed in the control cells. Interesting results of transcriptional activity were also obtained for the extranucleolar chromatin. As compared to the control group of cell nuclei, a 24 h HU treatment resulted in a significant reduction in transcription levels. Subsequent time points of incubation with HU (48 and 72 h treatments) caused an increase in EU fluorescence intensity, still lower; however, than in the series of untreated root meristem cells. Therefore, these results may suggest that NuS becomes triggered under specific conditions, such as transcriptional dysregulation or DNA damage. This reaction is usually characterized by changes in the size and morphology of nucleoli or uncontrolled rDNA transcription [45]. It is also accompanied by the redistribution or degradation of key nuclear factors, including fibrillarin (FBL) and nucleophosmin (NPM) [45]. These results may provide a basis for further research on RS and NuS in plants and for the search for correlations between these two types of stress.

The correct course of DNA replication is crucial to maintaining the integrity of the genome. Any abnormality in this process, e.g., such as that exerted by HU-induced deficit of deoxyribonucleotides, inevitably leads to RS (Figure 9). Many repetitive sequences in nuclear DNA are potentially susceptible to RS, as they often generate secondary DNA structures capable of interfering with the DNA replication process. Many copies of ribosomal genes (rDNA) present in the nucleus represent such RS-prone regions [46]. Reducing the rDNA copy number has been shown to stop the excessive accumulation of blocked RFs, thus allowing cells to survive and adapt to RS [47,48].

## 4. Materials and Methods

### 4.1. Plant Material

Sterile seeds of onion (*Allium cepa* L.) were sown in large Petri dishes lined with moist tissue paper and left in the dark at 24 °C. After 96 h of germination, seedlings with roots of about 1.5 cm in length were transferred to Petri dishes containing distilled water (control) and 0.75 mM HU solution, respectively. Incubation periods (24, 48, and 72 h) and concentration of HU (0.75 mM) were chosen consistent with previous tests on root meristem cells of *A. cepa* [18,49].

### 4.2. DNA Staining by the Feulgen Method

Feulgen staining was performed according to Żabka et al. [17]. Onion roots were fixed for one hour in Carnoy’s mixture (absolute ethanol and glacial acetic acid; 3:1, *v*/*v*). After washing with ethanol (3 × 10 min), cut root tips were rehydrated (3 × 20 min) and hydrolysed in 4M HCl. After 60 min, the material was stained with Schiff’s reagent (Sigma–Aldrich, St. Louis, MO, USA). After washing three times with SO_2_ and distilled water, root meristems were squashed on basal slides in a drop of 45% acetic acid. Slides were frozen on a dry ice surface, rinsed in 70% alcohol, dried, and embedded in Canada balsam.

### 4.3. Immunocytochemical Detection of Fibrillarin (FBL)

The procedure for immunofluorescent detection of fibrillarin (FBL) was adapted from Żabka et al. [17]. Root meristems from control plants and those treated for 24, 48, and 72 h with 0.75 mM HU were fixed for 45 min (room temperature) in PBS-buffered 4% paraformaldehyde solution. The material was then rinsed three times with PBS and placed in a citric acid-buffered digestion solution (pH 5.0) containing 2.5% pectinase, 2.5% cellulase, and 2.5% pectolysin (37 °C). After 20 min, root tips were washed 2 times with PBS (#P4417, Merck, Darmstadt, Germany), placed on Super Frost Plus slides (Menzel–Gläser, Braunschweig, Germany) in a drop of water, and crushed. After drying, the dissociated cells were pre-treated with PBS-buffered 8% BSA and 0.1% Triton X-100 (#T9284, Merck) for 50 min (room temperature) and incubated with monoclonal anti-FBL antibody produced in mouse (1:400 dilution; # WH0002091M1, Merck), dissolved in PBS containing 1% BSA. Antibody dilutions were optimized, and negative controls (without primary antibody) were performed. Incubation was carried out in a humidified atmosphere (4 °C) for 24 h. After rinsing with PBS, slides were incubated for 1.5 h (room temperature) with Alexa Fluor^®^488-conjugated goat anti-mouse IgG secondary antibodies (1:400; #4408, Cell Signaling, Danvers, MA, USA) and stained with PI. Samples were embedded in a mixture of PBS/glycerol (9:1) with 2.3% DABCO.

### 4.4. 5-Ethynyl Uridine Incorporation

Detection of nascent RNA was performed according to Żabka et al. [17]. Control and 0.75 mM HU-treated *A. cepa* seedlings were incubated for 60 min, in the dark, in 1 mM 5-ethynyl uridine solution (5-EU; Thermo Fisher Scientific, Waltham, MA, USA; the control series was prepared without 5-EU). To verify the specificity of 5-EU incorporation, a negative control was made using RNase. Cut meristems were fixed in 4% PBS-buffered paraformaldehyde (4 °C; pH 7.4) for 45 min, washed three times with PBS, and macerated for 45 min with 2.5% citrate-buffered pectinase (pH 5.0; 40 °C). After washing two times with PBS, meristems were squashed on slides (Polysine™, Menzel–Gläser) in a drop of distilled water. After freezing on a dry ice surface, the coverslips were removed with a sharp razor blade, and the slides were washed with water and air-dried. Cells spread on the slide surface were permeabilised with 0.5% Triton X-100 for 20 min. RNA was visualized using the Click-iT^®^ RNA Alexa Fluor^®^488 Imaging Kit with reaction cocktail (Thermo Fisher Scientific, Waltham, MA, USA). After 60 min incubation (room temperature), slides were washed in Click-iT^®^ wash buffer (Thermo Fisher Scientific, Waltham, MA, USA) and PBS, stained for 1 min with propidium iodide (PI; 0.3 mg mL^−1^), and rinsed in PBS. Next, they were washed with PBS and embedded in a mixture of PBS/glycerol (9:1) with 2.3% DABCO.

### 4.5. Observations and Microscope Analyses

Observations of cells stained with 5-EU and with antibodies against FBL were made under a Nikon Eclipse E600W fluorescence microscope (Tokyo, Japan) equipped with a B2 filter (blue light; *λ* = 465–496 nm) for Alexa Fluor^®^488, or a G2 filter (green light; *λ* = 540/25 nm) for propidium iodide (PI)-stained cell nuclei. All images were recorded at the same time of integration using a DS-Fi1 CCD camera (Nikon, Tokyo, Japan). Quantitative analyses and nuclear DNA fluorescence measurements were made after converting color images into greyscale and expressed in arbitrary units as mean pixel value (pv) spanning the range from 0 (dark) to 255 (white) according to methods described earlier [17,18]. Feulgen-stained cell nuclei were photographed using a Nikon Eclipse E600W microscope (Tokyo, Japan).

In order to obtain the percentage of cells with aberrations (aberration index) after 72 h of incubation in HU, 100 cells (stained using the Feulgen method) from each of the four root tips were analyzed. Three biological replicates were performed.

All fluorescence assays were repeated at least four times, and the data set contained 100 cells per biological replicate.

The size threshold used to define CBs is from 1 to 2 µm.

Fluorescence intensity analyses, the line plots, and interactive 3D Surface Plots were performed using the ImageJ software ver. 1.54p.

### 4.6. Statistical Analyses

Statistical analyses and outcomes visualization were performed in GraphPad Prism 10 software. Student’s *t*-test was used to determine the difference between the averages of two independent groups. Statistical significance, denoted by asterisks (*), was attributed to *p*-values ≤ 0.05. The data obtained from all experiments were expressed as mean values ± standard deviation of the mean (±SD).

## 5. Conclusions

Our research shows that 24-, 48-, and 72 h exposure to 0.75 mM HU induces RS in root meristem cells of *Allium cepa*. The prolonged stress resulted in a ~2-fold increase in pre-rRNA transcription level after 24 h of incubation in HU, followed by a gradual decrease in 5-EU fluorescence. Both in control material and during successive periods of incubation with HU, a highly conserved nucleolar protein, FBL, accumulated in FCs, DFC, and GC. In addition, our analyses showed that after 48 and 72 h of HU treatment, the number of CBs was almost twice as high as that observed in the control cell nuclei. In summary, our results may suggest that prolonged replication stress may contribute to nuclear changes and induce nucleolar stress.

## Figures and Tables

**Figure 1 ijms-26-11321-f001:**
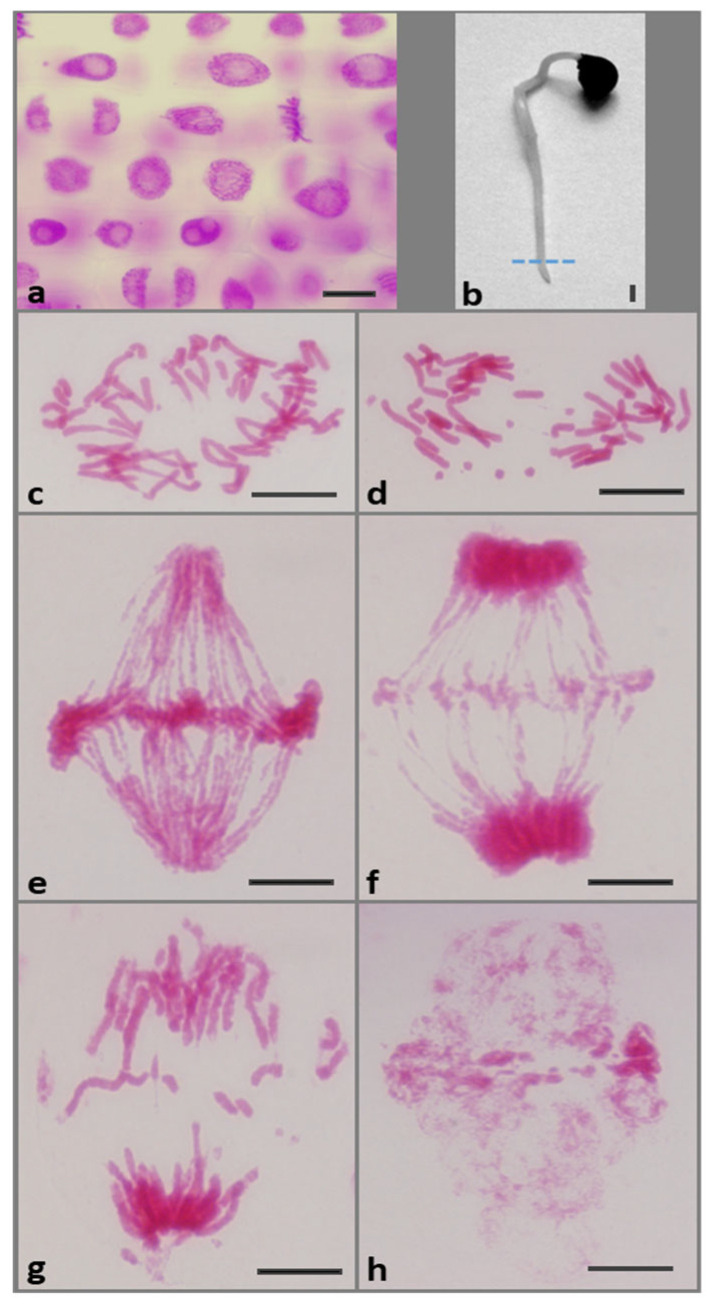
(**a**) Feulgen-stained cell nuclei from (**b**) root meristem of *A. cepa* seedlings incubated in water (control). The dotted line shows the boundary between the meristem (M) and the elongation zone (EZ). Scale bars = 10 μm and 10 mm, respectively. Symptoms of chromosome aberrations after 72 h treatment with 0.75 mm HU during: (**c**,**d**) early anaphase, (**e**–**g**) ana-telophase; (**h**) post-telophase cell nuclei with micronuclei. Scale bar = 20 μm.

**Figure 2 ijms-26-11321-f002:**
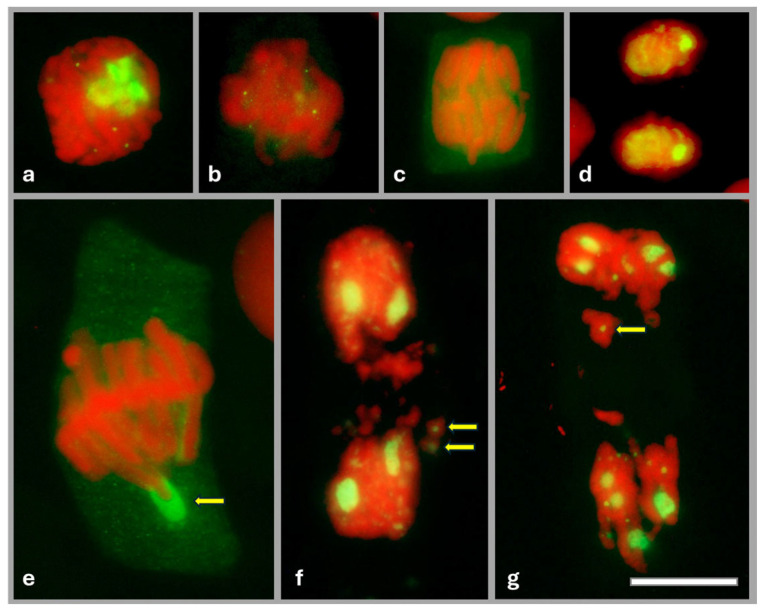
Location of FBL using anti-FBL antibodies and secondary antibodies conjugated with Alexa Fluor^®^488 (green) in meristematic cells: the control (**a**–**d**), 72 h treatment of 0.75 mM HU (**e**,**g**). Prophase (**a**), metaphase (**b**), anaphase (**c**), telophase (**d**), metaphase (**e**), yellow arrow indicates location of the FBL around the terminal parts of the chromosome, and telophase (**f**,**g**); yellow arrows indicate the location of FBL in the fragmented chromosomes. DNA stained with propidium iodide (PI; red). Scale bar = 20 μm.

**Figure 3 ijms-26-11321-f003:**
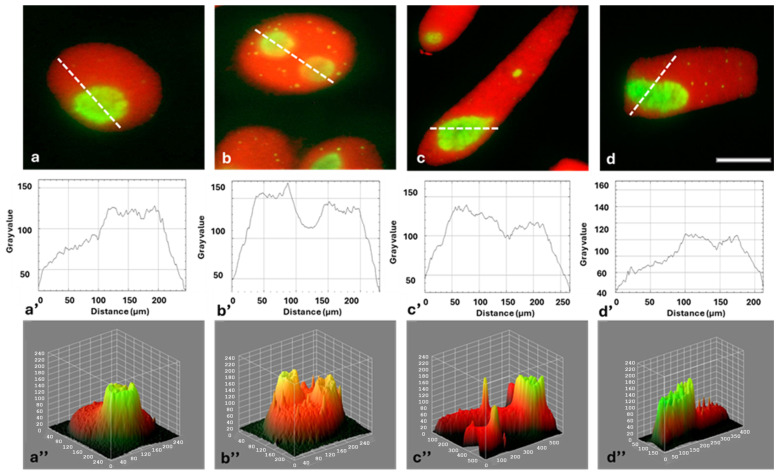
Detection of FBL immunofluorescence in root meristem cell nuclei (G2 phase) of *Allium cepa*: control (**a**), after 24 h (**b**), 48 h (**c**), and 72 h treatment with HU (**d**). Scale bar = 20 μm. DNA stained with PI (red). Corresponding densitometric plots showing changes in the intensity of fluorescence of the nuclear and nucleolar regions in: the control (**a’**) and after: 24 h (**b’**), 48 h (**c’**), and 72 h treatment with HU (**d’**); the course of each plot is marked by a white dashed line in (**a**–**d**). Interactive 3D surface plots: control (**a”**), 24 h (**b”**), 48 h (**c”**), and 72 h treatment with HU (**d”**).

**Figure 4 ijms-26-11321-f004:**
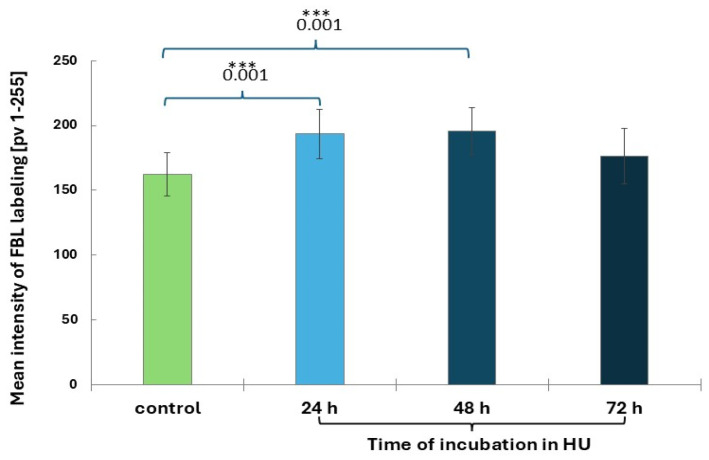
Mean intensity of FBL labeling in the control and 0.75 mM HU-treated root meristem cells. Error bars represent standard deviation (SD). When compared with the control, statistically significant changes in mean values (±SD) are marked by asterisks: *** indicates *p* < 0.001. Each mean value is based on the analysis of ten root meristems (N = 100 cells/meristem).

**Figure 5 ijms-26-11321-f005:**
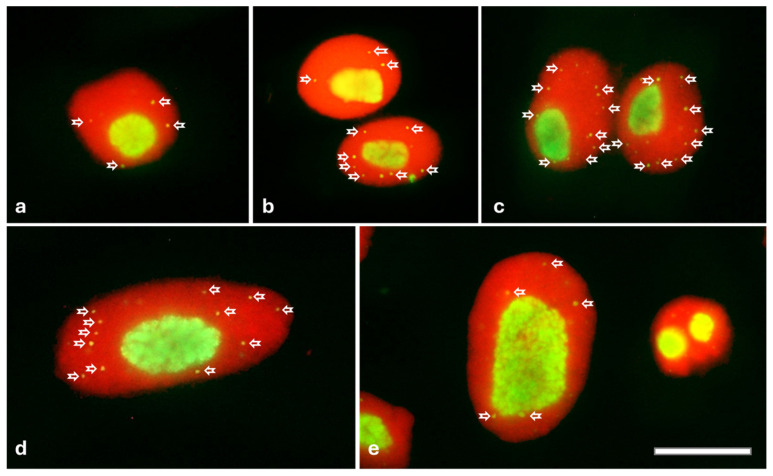
Immunofluorescence detection of FBL in Cajal bodies (CBs; indicated by white arrows) in cell nuclei during the G2 phase: the control (**a**), after 24 h (**b**), 48 h (**c**), and 72 h treatment with HU (**d**,**e**). Scale bar = 20 μm.

**Figure 6 ijms-26-11321-f006:**
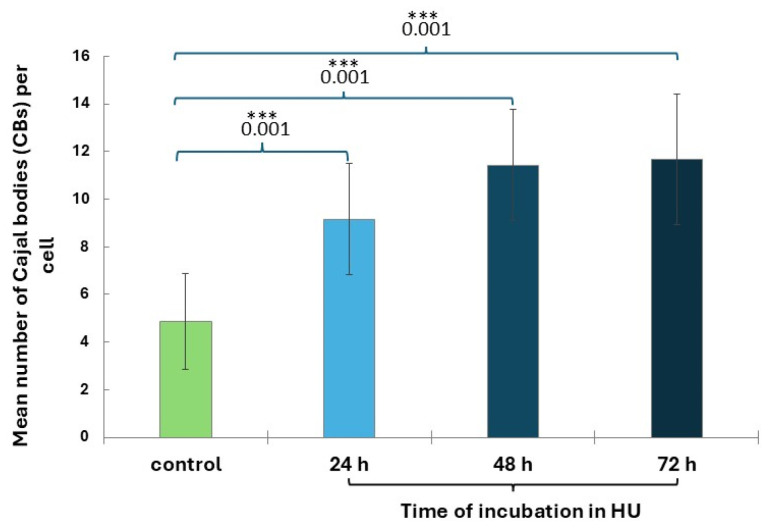
Mean number of ajal bodies (CBs) in the control and 0.75 mM HU-treated root meristem cells. Error bars represent standard deviation (SD). When compared with the control, statistically significant changes in mean values (±SD) are marked by asterisks: *** indicates *p* < 0.001. Each mean value is based on the analysis of ten root meristems (N = 100 cells/meristem).

**Figure 7 ijms-26-11321-f007:**
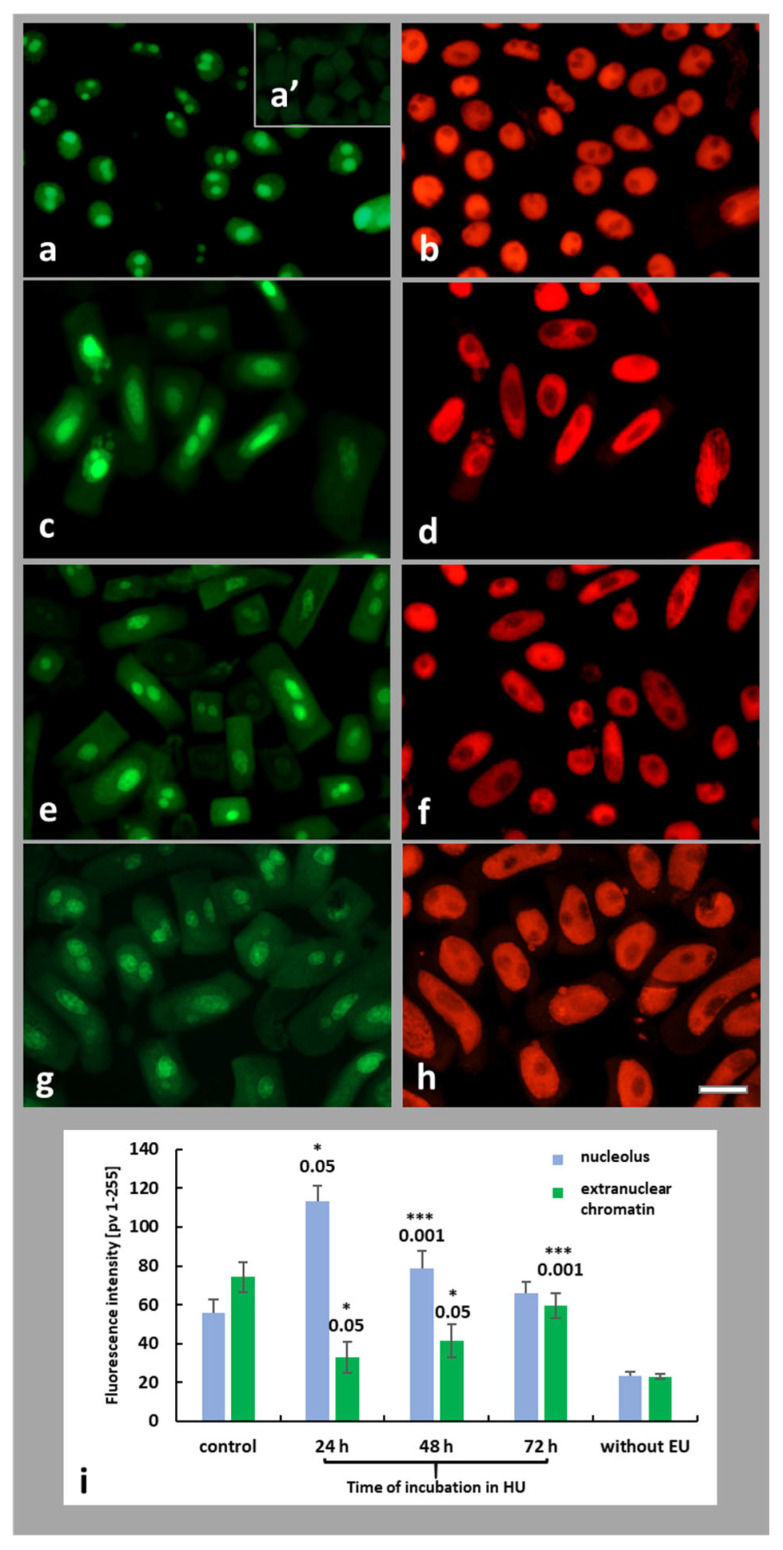
Fluorescence intensity of the extranucleolar chromatin and the nucleoli following application of the EU staining method: control (**a**), 24 h (**c**), 48 h (**e**), and 72 h HU treatment (**g**); corresponding images of the same cells counterstained with PI (**b**,**d**,**f**,**h**). Weak fluorescence signal obtained using experimental procedure without EU (**a’**). Scale bar  =  20 μm. Microfluorimetric evaluation of mean (total) nascent transcription levels measured for the nucleolus (blue) and extranucleolar chromatin (green) in the control and HU-treated onion root meristem cells (**i**). When compared with the control, statistically significant changes in mean values (± SD) are marked by asterisks: * indicates *p* < 0.05 and *** indicates *p* < 0.001. Each mean value is based on the analysis of ten root meristems (N = 100 cells/meristem).

**Figure 8 ijms-26-11321-f008:**
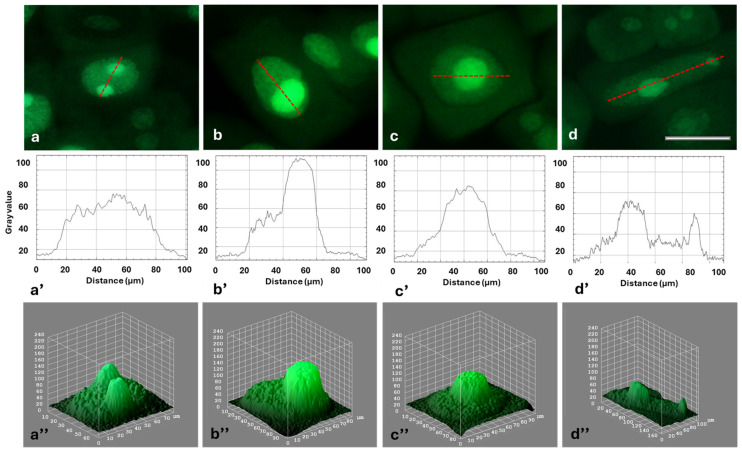
Fluorescence detection of EU incorporation into the G2 phase root meristem cell nuclei of *Allium cepa*: control (**a**), 24 h (**b**), 48 h (**c**), and 72 h treatment with HU (**d**). Scale bar = 20 μm. Densitometric plots showing changes in the intensity of fluorescence of the extranucleolar chromatin and nucleolar regions in the control (**a’**) and after successive HU treatments (**b’**–**d’**). The course of each plot is marked by a red dashed line. Interactive 3D surface plots: control (**a”**), 24 h (**b”**), 48 h (**c”**), and 72 h treatment with HU (**d”**).

**Figure 9 ijms-26-11321-f009:**
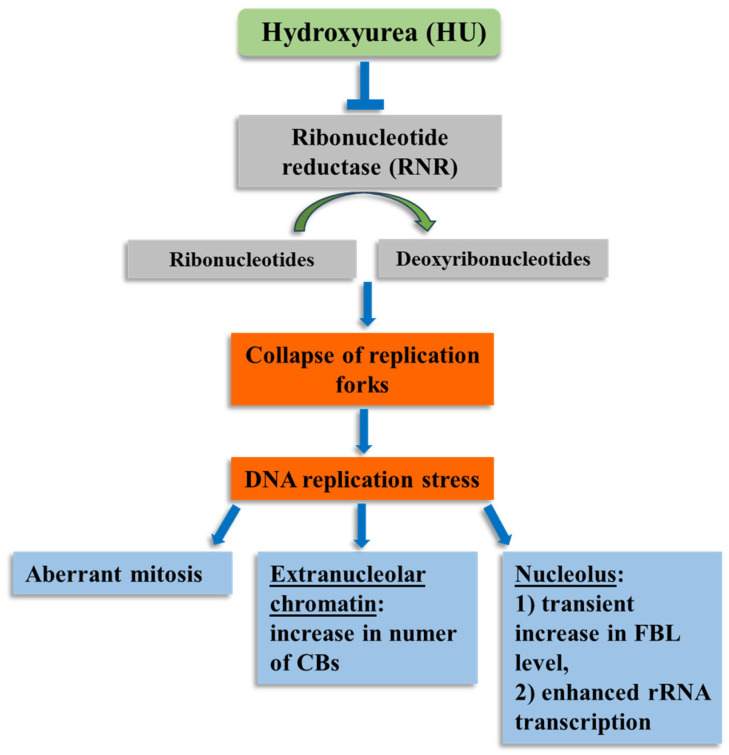
Chosen effects of HU on root meristem cell nuclei of *A. cepa*.

## Data Availability

The original contributions presented in this study are included in the article. Further inquiries can be directed to the corresponding author.

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
