# Peer review of "Identification of Fibrillarin and Cajal Bodies Under DNA Replication Stress Conditions in Root Meristem Cells of Allium cepa"

_ijms, 2025, doi:10.3390/ijms262311321_

Round 1

Reviewer 1 Report

Comments and Suggestions for Authors

In this work, the authors aimed to investigate the impact of hydroxyurea (HU) treatment on the nucleolar protein fibrillarin (FBL) localization and rRNA transcription levels in meristematic Allium cepa root cells. The content of this study is interesting. However, the whole manuscript is not well presented, which must be improved.

Allium cepa in the title should be in italic.

The language of the whole manuscript should be improved.

Abstract: The aim of this study should be clearly introduced in the abstract. Some abbreviations are not explained, such as RNP.

Introduction: This section should be carefully modified. The authors have introduced too much general information in the first three paragraphs. However, the main contents of this work is only briefly introduced in the last paragraph, which should be greatly extended.

Figure 1: Scale bar should be given for each picture.

Discussion: Many background information of this work should be placed in the introduction, such as lines 197-220.

Based on the results of this work, could the authors give a diagram to illustrate the action of mechanism of HU on Allium cepa root cells?

Materials and Methods: Proper references should be given for each method.

The conclusion is not found in the main text.

Reviewer 2 Report

Comments and Suggestions for Authors

Allium cepa in italics. Verify throughout the text.

Restructure the abstract to include: objective, a brief description of what was tested and analyzed, main results, and biological implications.

Remove keywords from the title. The title may already be used to search for similar research, so it is better to use different words in this section to broaden search results.

Introduction:

Check the citation style; Vancouver style citations should be enclosed in [ ]

Very long paragraphs should be broken up.

Include a brief justification for the use of Allium cepa in the research.

What is the study hypothesis?

What is the research objective?

Results:

Presenting the results before the methodology is interesting. However, this makes it difficult to understand what was researched and analyzed.

Organize the structure so that there are no large spaces between figures and information.

Improve the resolution of figures and images.

Materials and methods:

Insert citations and references for the methods used in the research.

Insert brand, model, city, and country of manufacture of the equipment used.

The analysis of the data obtained needs to be more detailed.

Reviewer 3 Report

Comments and Suggestions for Authors

Lines 12–22 assert several findings (reduced rRNA transcription, nucleolar stress, FBL redistribution, increase in CBs). Please (a) quantify the main effects in the abstract (e.g., “~2-fold increase in nucleolar EU signal at 24 h”), (b) avoid causal overstatements (e.g., “ultimately leads to induction of chromosomal aberrations” — replication stress contributes to aberrations; be cautious about causal wording), and (c) make the primary conclusions concise.

Introduction-Lines 26–38 provide a long description of RS and replication forks; some sentences are redundant. Suggest shortening and focusing on (i) HU as a model RS inducer, (ii) rationale for studying FBL/CBs in plants, and (iii) the knowledge gap. Also check and standardize citation formatting (some recent refs are 2023–2024; ensure style matches journal).

Line 205–206: “72 h incubation of 4-day-old primary roots of A. cepa (22, 23, 24, 25, 26) and Vicia faba (27) with 0. 75 mM HU” — there is an extra space after “0.” Please correct to “0.75 mM HU”. This spacing error appears in multiple locations (search manuscript and standardize to “0.75 mM”).
In Figure 1 legend (lines 100–105) show representative images but do not report how many cells/roots were analyzed to generate the percentage in the main text. Please add: number of biological replicates (seedlings/Petri dishes), number of meristems/crushed root tips per condition, and whether the images are representative. Also specify the imaging magnification and whether images were contrast-adjusted.

Lines 356–364 state “Immunofluorescence assays were repeated at least four times and the total number of analysed cells for one data set was always 100.” Please clarify: does “one data set” = 100 cells per biological replicate, or 100 cells pooled across replicates? For each quantified plot (Figures 4, 6, 7, etc.) provide exact N (biological replicates and total cells), the statistical test used (you say Student’s t-test — justify its use; were data checked for normality? If comparisons involve >2 groups, consider ANOVA with post-hoc tests). Add a statistics subsection specifying α, software, and details.

Lines 301–308: state 20 crushed root tips per time point were used for I–M cells. For other assays (FBL IF, 5-EU), specify how many independent seedlings/roots were used per treatment (biological replicates) in the Methods, not just in the Results.

Lines 338–355 (5-EU): indicate whether EU uptake was equal across treatments (e.g., did HU affect cell permeability or EU uptake?). State whether a transcription inhibitor negative control (e.g., actinomycin D) or RNase treatment was used to validate specificity of the EU signal.

Lines 93–99: “Percentage of cells with chromatin aberrations reached a limit of about 15% after 72 h incubation in HU.” What does “reached a limit” mean? Replace with precise language (e.g., “the frequency of cells with chromosomal aberrations increased to ~15% at 72 h”). Also add SEM or SD and N for that percentage. If possible, include statistical comparison versus control.

Lines 112–121 describe altered FBL localization at metaphase/telophase. Please quantify: how many metaphase or telophase cells showed peritelomeric FBL localization or FBL associated with fragments (report counts or percentages and N). Replace qualitative wording (“in some metaphase cells”) with numbers.
Figure 4 legend indicates Student’s t-test and *** p < 0.001. Please add (a) what comparisons the asterisks represent (control vs each treatment or pairwise), (b) sample sizes (n), and (c) whether tests were one- or two-tailed. If multiple pairwise comparisons were made, adjust for multiple testing. 

Lines 152–155 and Fig.6 report ~2–2.5× increase in CB abundance. Please (a) explain how CBs were detected and counted (manual vs automated), (b) clarify size threshold used to define a CB, (c) provide absolute mean ± SD (not only fold change), and (d) indicate whether increases were statistically significant (provide p-values and N).

The 2-fold increase in nascent transcription at 24 h followed by decrease (lines 173–175, 235–238) is interesting but requires more cautious interpretation. Could the transient increase be due to stress-induced rDNA hyperactivation or to a compensatory increase in nucleolar activity in surviving cells? Add a short discussion (and possibly additional controls) addressing alternative explanations (e.g., differential cell-cycle distribution, cell death, EU uptake differences). If feasible, present nucleolar vs extranucleolar EU quantification side-by-side and indicate whether total cellular transcription changed.

Lines 320–336 describe FBL immunostaining. Please add the following: catalog numbers and host species of primary antibody (you give “Sigma-aldrich” but include exact product code), catalog number for secondary antibody, whether secondary was pre-adsorbed, blocking conditions (you list BSA and Triton — specify buffer composition), and the exact mounting medium composition (DABCO % already given later, but standardize). Also indicate whether antibody dilutions were optimized and whether negative controls (no primary) were performed.

Lines 238–241 and 286–293 suggest HU “ultimately may contribute to the reduction of pre-rRNA transcription and induction of NuS” and CBs are “transcription-dependent ... responsive to RS.” Tone this down where evidence is correlative. Rephrase to emphasize correlation and propose mechanisms as hypotheses for future work. If claiming causation, include additional experiments (e.g., rescue by nucleoside supplementation or using RNR-independent RS inducers) or clearly label as speculative.

Line 371–372: “Data will be made available upon request.” Many journals require deposition of raw image data/quantification in a public repository; please either deposit data (FI measurements, raw images, counting spreadsheets) in a public repository (with accession link in the manuscript) or provide a concrete statement about how data will be shared (and include DOIs).

Check references for formatting consistency (e.g., missing parentheses or extra spaces around years in several entries). There are typos like “Rakitina, D.V.; Taliansky, M,.; Brown” (comma error)-please carefully proof the reference list.

Comments on the Quality of English Language

The manuscript has useful data but contains numerous grammatical slips, duplicated words, and awkward phrasing (e.g., “differences, in the localization” extra comma). Please perform a careful language edit (native English proofreading) throughout to improve clarity and flow.
